# Non-Gestational Ovarian Choriocarcinoma: A Rare Ovarian Cancer Subtype

**DOI:** 10.3390/diagnostics12030560

**Published:** 2022-02-22

**Authors:** Sean Cronin, Nishat Ahmed, Amaranta D. Craig, Stephanie King, Min Huang, Christina S. Chu, Gina M. Mantia-Smaldone

**Affiliations:** 1Department of Obstetrics and Gynecology, Albert Einstein Medical Center, Philadelphia, PA 19141, USA; AhmedN01@einstein.edu; 2Department of Surgical Oncology, Division of Gynecology Oncology, Fox Chase Cancer Center, Philadelphia, PA 19111, USA; amaranta.craig@tuhs.temple.edu (A.D.C.); stephanie.king@fccc.edu (S.K.); christina.chu@fccc.edu (C.S.C.); gina.mantia-smaldone@fccc.edu (G.M.M.-S.); 3Department of Pathology, Fox Chase Cancer Center, Philadelphia, PA 19111, USA; min.huang@fccc.edu

**Keywords:** non-gestational ovarian choriocarcinoma, ovarian tumor, germ cell tumors, rare ovarian tumors

## Abstract

Non-Gestational Ovarian Choriocarcinoma (NGOC) is an extremely rare ovarian tumor, with an incidence of less than 0.6% of malignant ovarian germ cell tumors. Its close pathologic resemblance to Gestational Ovarian Choriocarcinoma (GOC), however, requires special attention as the treatments differ greatly. NGOC typically affects patients in late adolescence or early reproductive years. As a result, NGOCs are often misdiagnosed as ectopic pregnancies due to their common presentation of bleeding, abdominal pain, adnexal mass, and positive serum beta-HCG. On pathologic examination, the tumor is indistinguishable from GOC, and only after review of tissue for paternal genetic components can the diagnosis of NGOC be made. Imaging studies often show highly vascular lesions with further investigation with computer topography (CT) sometimes showing metastatic lesions in the lungs, pelvis, vagina, and liver. These lesions are often hemorrhagic and can lead to catastrophic bleeding. Treatment is vastly different from GOC; NGOC requires treatment with both surgical resection and chemotherapy, with Bleomycin, Etoposide, and Cisplatin (BEP) being the most used regimen. With correct diagnosis and treatment, patients can often receive fertility sparing treatment with long term survival.

## 1. Introduction

Ovarian choriocarcinoma is an extremely rare form of ovarian cancer. It can be broadly classified into two variants, gestational ovarian choriocarcinoma (GOC) and non-gestational ovarian choriocarcinoma (NGOC). NGOC are further subdivided into mixed, which contain other germ cell components, and pure subtypes, which contain only choriocarcinoma. The incidence of GOC is 1:369,000,000 whereas the incidence of NGOC accounts for just 0.6% of malignant ovarian germ cell tumors [1,2,3,4,5]. GOC is a form of gestational choriocarcinoma and related to a patient’s previous pregnancy history and may exist concurrently with a well-developed corpus luteum, with cure rates approaching 90% with single agent chemotherapy, typically methotrexate [6]. 

NGOCs are unrelated to pregnancy, and DNA analysis demonstrates the absence of any paternal genes [7,8,9]. NGOCs often occur in children and young adults, arising from midline structures that form during embryogenesis or primordial germ cells in the gonads after birth and demonstrate trophoblastic differentiation [7,10,11]. It has also been proposed that NGOCs arise from “retrodifferentiation” to an earlier embryonic cell stage of somatic tumors that have already undergone neoplastic transformation [12]. 

NGOCs are characterized by rapid growth and a relatively poor prognosis; overall survival of International Federation of Gynecology and Obstetrics (FIGO) stage I, II, and III disease is 100% over 3 years, with the survival rate of FIGO IV disease dropping to just 25% at 3 years. When divided into pure and mixed NGOC tumors, the former have a 94% overall survival while the latter have just 50% overall survival at 3 years [13].

NGOC is often not considered in the initial differential diagnosis of an adnexal mass. However, obtaining a detailed history can help direct the clinician to correctly identifying cases. The single most important part of the patient’s history to differentiate between NGOC and GOC is a history of previous pregnancy. If correctly diagnosed and treated there is still the possibility for favorable outcomes, with regards to both overall survival and fertility preservation. In this review, we will explore the background, presentation, and described treatment protocols of NGOC.

## 2. Background

There are two types of NGOC: the pure type and the mixed type. The pure type, which is extremely rare, only contains choriocarcinoma [14]; the absence of other germ cell elements is demonstrated by the lack of immunohistochemical staining for CD30, PLAP, and AFP [15]. The mixed type contains other germ cell tumors, including immature teratomas, endodermal sinus tumors, embryonal carcinomas, and dysgerminomas [13]. Histologically, both GOC and NGOCs have identical presentation with abnormal trophoblastic hyperplasia and anaplasia, absence of chorionic villi, high proliferative index, and the presence of hemorrhage and necrosis within the tumor tissue [2,16]. The histologic similarities between GOC and NGOC contribute to the diagnostic conundrum, see Table 1 [2,14,17]. One study compared the genetic molecular biology of GOC and NGOC, and found mutations involving *DNAJB9*, a negative feedback regulator of *p53* and NGOC cells showing aberrant expression of *p53* [18]. Genetically engineered mouse models with alterations in *Trp53* gene were also shown to develop NGOC [19]. NGOC cells have also demonstrated copy number variations and significant amplification of *Her2*, *IKZF3*, *PGAP3*, and *C-Myc*, which are not demonstrated by GOC; these genes have been implicated in the poorer immunogenicity of NGOC, thus resulting in less sensitivity to chemotherapy [12]. Another study demonstrated gain of 21p11, which has also been observed in other somatic tumors and germ cell tumors [20,21].

## 3. Presentation and Diagnosis

The symptoms of NGOC are vague and non-specific, and the age at presentation aligns with other beta-HCG producing conditions, including ectopic pregnancy. NGOC usually affects younger reproductive-aged woman and often presents with metastatic disease [22]. In the largest case review done by Lui et al., 39 case reports were reviewed with the peak age of onset ranging 12 to 25 years [13]. The most common signs and symptoms are vaginal bleeding, abdominal pain, adnexal mass on ultrasound, and a positive pregnancy test [6,11,13,14,22]. In children, initial presentation may also include precocious puberty [23]. Most cases described in the literature involved unilateral masses, but one case report describes an extremely rare case of bilateral NGOC [24]. Since NGOCs are typically unilateral, cases of NGOCs are often initially misdiagnosed as ectopic pregnancy [4,13,25,26,27].

### 3.1. Clinical

Patients with non-specific abdominal pain, bleeding, and a positive pregnancy test are often evaluated in the emergency department with a transvaginal ultrasound which yields non-specific findings. Pelvic ultrasound often shows a highly vascular, echogenic non-homogeneous unilateral mass and a normal uterus with a thin endometrial stripe [3,24,28]. Follow up imaging with Computed Tomography (CT) is then usually obtained. CT images can further help to evaluate the extent of the disease and the presence of hemorrhagic lesions in other locations [3,13,29].

Similar to GOC, hemorrhage causes significant morbidity and mortality in patients with NGOC [3]. The vascularity is due to the trophoblastic cells’ innate capacity to invade and erode vascular structures [13]. Bleeding is the most often reported presentation of NGOC [30], including non-gynecologic bleeding such as profound hematochezia in a patient who had metastatic disease involving the intestines [22]. As a result, patients often have severe anemia requiring multiple blood transfusions or massive transfusion protocols [22,31]. In one case report, a patient received 19 blood transfusions in the year prior to her diagnosis [3]. 

The extremely high levels of beta-HCG may trigger ovarian hyperstimulation syndrome. Other clinical manifestations, as a result of the increased vascular permeability mediated by vasoactive agents produced by the corpora lutea, include pleural or pericardial effusion, electrolyte imbalances, hemodynamic instability, and coagulopathy [5].

### 3.2. Laboratory

Beta-HCG levels for NGOC patients are often extremely elevated, as is typically seen in GOC. At beta-HCG levels seen in NGOC cases, the Hook Effect becomes relevant [32]. When using point of care urine pregnancy tests, the Hook Effect occurs in sandwich immunoassays when the antigen concentration is high enough to saturate both the migratory phase and fixed detection antibodies independently, rather than binding occurring to subunits of the same molecule [14,15]. Thus, a falsely negative test could potentially lead to a delay in diagnosis and allow for progression of disease [33]. This can be overcome by diluting the sample, either 1:10 or 1:100, to allow for dilution of the beta-HCG [34]. However, serum-based beta-HCG tests are not subject to the Hook Effect and will reveal the true elevation of beta-HCG, which has been reported to be as high as 1 million [14]. 

### 3.3. Genetic Testing

Previously, the primary way to diagnose patients with NGOC was with a detailed patient history, including recent sexual activity and antecedent pregnancy. If there is no history of intercourse and/or an antecedent pregnancy, then a patient would be diagnosed with NGOC [14,16,26,35]. In 1963, Saito and colleagues developed a 4-item set of diagnostic criteria for NGOC: (1) absence of disease in the uterine cavity, (2) pathological confirmation of choriocarcinoma with the persistence of elevation in beta-HCG, (3) exclusion of molar pregnancy, and (4) exclusion of coexisting intrauterine pregnancy [6]. 

With the availability of genetic tumor testing and the ability to identify paternal and maternal genetic components we can better diagnose NGOC. The presence paternal genetic material in the tumor tissue additionally helps to distinguish GOC from NGOC. In 2006, a small study examined tumor microsatellite DNA from 6 patients to identify tumor alleles that were not present in maternal tissue [35,36]. The presence of non-maternal alleles is diagnostic of GOC [2,4,8,35]. If paternal DNA is present, then the diagnosis of NGOC can be excluded [2,9,14,35,37]. Use of DNA short tandem repeat and polymorphism analyses has also been utilized to differentiate pure NGOC from GOC by revealing post-meiotic germ cell derivation [35,38,39,40,41]. One study was able to use fluorescence in-situ hybridization (FISH) analysis to differentiate GOC from NGOC by identifying paternal Y chromosome centromeres [42]. The presence of b2-microglobulin mRNA has also been used to distinguish GOC from NGOC [43]. Given that these two diseases have different therapeutic and prognostic outcomes, a confirmatory diagnostic assay is important at the onset of the disease. 

### 3.4. Staging

The aggressive nature of NGOC makes metastatic disease a concern, with hematologic and local spread occurring early in the disease process [44]. Local spread appears to follow the embryological pathway of germ cell migration [45]. In the largest chart review of 39 patients with NGOC, 80% of patients had metastatic disease to the lungs, 30% to the pelvis, 20% to the vagina, and 10% to the liver [13,46]. Other sites with less common metastatic disease included the gastrointestinal tract, spleen, and kidney [3,4,13,16,22]. Cerebral metastases have also been identified and one case report also described delayed diagnosis of brain metastases discovered two years after primary treatment [1,3]. 

Staging for NGOC remains unclear. In a review article by Shao et al. looking at 37 patients with NGOC, stage was calculated using both the 2013 FIGO staging for ovarian cancer and the 2000 FIGO staging for choriocarcinoma. In the 2013 staging scheme for ovarian cancer, the distribution of stage I was 41.2%, stage II 5.9%, stage III 5.9%, and stage IV 47.1% [6]. Using the FIGO 2010 staging scheme for choriocarcinoma, the distribution was stage I 41.2%, stage II 33.4%, and stage III 26.5%. These distributions were again observed in another case series by Lui et al. of 39 patients with choriocarcinoma; however, they only reported the staging using the staging for ovarian cancer [13]. 

## 4. Treatment

Due to the rarity of pure NGOC, there have not been large-scale studies evaluating the optimal surgical treatments. Most published articles related to NGOC are case reports or case series with short discussions about the current evidence used for their treatment. Table 2 summarizes the findings of these case reports and case series to show current practices used to treat this extremely rare condition. 

### 4.1. Surgery

The basis for surgical resection is extrapolated from the treatment of germ cell tumors. Shao et al. reported on patients who underwent either cytoreductive surgery (resection of the uterus, bilateral ovaries, bilateral fallopian tubes, omentum, pelvic and para-aortic lymph nodes, appendix, and any other abdominal, pelvic metastases) or fertility-preserving surgery (any combination of removal of tumor and reproductive organ not resulting in sterilization) [6]. Six patients who were initially treated with fertility-preserving surgery subsequently underwent cytoreductive surgery, most commonly due to an unsatisfactory decrease in beta-HCG or relapse. One patient who initially underwent fertility-sparing surgery subsequently underwent total hysterectomy to address profuse vaginal bleeding due to the mistaken diagnosis of primary uterine choriocarcinoma, but evaluation of the uterine and adnexal specimen showed no evidence of disease [27]. 

Liu et al. reported complete cytoreduction (R0) in 54% of reviewed cases [13]. In patients where R0 resection was obtained, there was an overall survival improvement at greater than 20 months. Complete cytoreduction was more easily obtained in patients with lower FIGO scores, based on the 2010 choriocarcinoma staging. However, even in patients with advanced disease (i.e., FIGO stage IV) who received chemotherapy prior to surgery, benefit was observed with surgical cytoreduction, with 3-year overall survival at 92% for R0 compared to 73% for patients who did not achieve R0 status [13].

### 4.2. Fertility-Sparing Surgery

Fertility-preservation surgery should be discussed with patients at the time of surgical management, given that the peak incidence of NGOC is during a woman’s early/peak reproductive years. Liu et al. recommend that patients with suspected stage I disease be offered fertility-sparing surgery [13]. However, many patients already have advanced disease at presentation [6,13,47]; as seen in larger case series, more than 50% present with stage II, II, or IV disease [6,13]. Minimally invasive surgical approaches to ovarian cancer have previously been described [36,59]. Xin et al. reported a patient with stage IIb NGOC treated with fertility-sparing resection via minimally invasive approach followed by adjuvant chemotherapy and achieved remission at nine months after therapy [55]. Inaba et al. reported a patient with stage III NGOC treated with fertility-sparing subtotal tumor resection followed by high-dose chemotherapy and achieved complete remission after eighteen months [60]. 

Much of the data regarding fertility sparing treatments have been extrapolated from Malignant Ovarian Germ Cell Tumors (MOGCT). Yang et al. examined 31 patients who underwent fertility sparing surgery for MOGCT and achieved 33 successful live births. There was no statistically significant difference with respect to either progression free survival or overall survival in patients undergoing fertility sparing surgery compared to those who underwent complete resection and staging. As would be expected, post-operative residual tumor size was an independent prognostic factor for both overall survival and progression free survival. Yang et al. conclude that fertility sparing surgery with adjuvant chemotherapy had little or no effect on prognosis or fertility [23]. Given the known gonadotoxicty of chemotherapeutic agents, co-administration of gonatropin-releasing hormone analogs has been described to preserve remaining follicles [55,61]. Overall, the majority of patients who undergo fertility-sparing surgery and combination chemotherapy have resumption of normal ovarian function and associated fertility [56,62].

### 4.3. Chemotherapy

GOCs are often treated with methotrexate-based chemotherapy regimens based on the FIGO score; single agent methotrexate is often used for patients with a FIGO score of less than 7 [63]. However, single agent chemotherapy is ineffective in patients with NGOCs [9]. NGOCs fall within the diagnostic realm of a germ cell tumors and should be addressed as such from both a treatment and prognostic standpoint. However, NGOCs are more difficult to treat [2] and have a worse prognosis compared to GOCs [1,13,64]. Treatment typically consists of both surgery and systemic chemotherapy [6]. However, due to the rare occurrence of the disease and a lack of clinical trials, a preferred chemotherapy regimen has not been established [13]. Both platinum-based and multi-agent methotrexate-based treatments have been utilized for treating NGOCs. Most recent case reports describe treatment with Bleomycin, Etoposide, and Cisplatin (BEP), which had shown excellent activity in other malignant germ cell tumors; however, successful responses have been observed with other regimens [6,13,65,66].

The most used treatment protocol for NGOC is a platinum-based regimen. The use of platinum-based chemotherapy is based on germ cell tumor studies [22]. However, in two NGOC case series, alternative treatment protocols were utilized. Liu et al. reported 14 patients who received BEP, 3 patients who received etoposide, methotrexate, echinomycin/vincristine, and cyclophosphamide (EMA-CO), 2 patients who received single agent methotrexate, and 4 patients who received a few other treatment protocols. Of these 14 patients, only one had disease progression. The overall effectiveness of BEP was found to be 93% [13]. See Table 3 for a detailed explanation treatments from Liu and Shao. 

Immunotherapy has also been proposed, using nivolumab to target PD-L1 overexpression of choriocarcinomatous cells; anti-tumor immunity was shown to be restored in lung cancers with choriocarcinomatous features [12].

### 4.4. Radiation

Radiation therapy is infrequently used for patients with NGOC. Chemoradiotherapy was recommended in one case series for patients with advanced disease followed by palliative surgical resection of any residual disease [13]. In another case report, a patient received radiation therapy after suspected diagnosis during an exploratory laparotomy. She only received 5 fractions before complications arose and radiation therapy was stopped; treatment was then converted to platinum-based chemotherapy with BEP [22]. Use of radiation therapy has also been described in cases of cerebral metastases [1]. Overall, with the limited data available, the role of radiation in the treatment for patients with NGOC may need to be examined on a case-by-case basis.

### 4.5. Choriocarcinoma Syndrome

Patients with advanced NGOC may also develop choriocarcinoma syndrome, a rare but potentially fatal complication that should be suspected in patients with high tumor burden, significant metastatic disease, and elevated tumor markers [47]. This syndrome has been described as occurring after initiation of chemotherapy, or spontaneously in advanced disease, and is thought to be related to tumor invasion of small blood vessels and subsequent hemorrhage [47,67]. Clinical manifestations most commonly include pulmonary hemorrhage and acute respiratory distress syndrome, as well as gastrointestinal hemorrhage, intra-hepatic and/or intra-abdominal hemorrhage, hemo- or pneumothorax, and cerebral hemorrhage [7,47,68]. Choriocarcinoma syndrome has a very poor clinical prognosis; initiating milder chemotherapy regimens and ensuring multimodal supportive therapy, timely and sequential intensive chemotherapy has been proposed to prevent and manage choriocarcinoma syndrome [47]. Modified regimens of BEP have been proposed to prevent this fatal syndrome and have shown favorable results [7,69]. 

## 5. Follow Up

Surveillance of NGOCs is also typically extrapolated from MOGCT. Beta-HCG monitoring is frequently utilized, with the inference that if the beta-HCG is negative then there is a low probability of residual/recurrent disease. Treatment success is documented after the patient’s beta-HCG levels normalize. Most case reports monitored the patients for 2 years with imaging after normalization of their beta-HCG [2]. Lui et al. laid out the most extensive follow up protocol. For their 37 cases they proceed with surveillance as follows [6]: 0–3 months—monthly serum quantitative beta-HCG with CT of the chest, abdomen, and pelvis;4–12 months—every 3 months serum quantitative beta-HCG with CT of the chest, abdomen, and pelvis;13–36 months—every 6 months serum quantitative beta-HCG with CT of the chest, abdomen, and pelvis;37–60 months—yearly serum quantitative beta HCG with CT of the chest/abdomen and pelvis;Serum quantitative beta HCG with CT of the chest, abdomen, and pelvis every 2 years thereafter.

Based on the limited data on follow up and case reports available, the above protocol seems to be the most conservative and will allow for close observation for possible recurrence of disease. In a clinical analysis of 21 patients, after a median follow up of 71.4 months, the overall survival was 79.4% [70].

## 6. Recommendations

Based on available published expert opinion and experience, we propose the following recommendations to help evaluate and guide treatment for patients diagnosed with NGOC. Due the extreme elevations in beta-HCG often seen in patients with NGOC, point-of-care urine pregnancy tests may be falsely negative and should not be relied upon to rule out an elevated beta-HCG. If ultrasound findings of a vascular adnexal mass are noted using an abdominal or vaginal approach, a serum-based beta-HCG should be obtained despite a previously negative urine pregnancy test (serum beta-HCG tests are not subject to the Hook Effect). Ectopic pregnancy can often be ruled out in NGOC patients due to 10–100 times higher beta-HCG. Beta-HCG is rarely greater than 100,000 mIU/mL in ectopic pregnancy. 

A comprehensive obstetric and gynecologic history should be obtained to distinguish between GOC and a NGOC. As discussed earlier, disease prior to first intercourse was previously used as a determining factor for the diagnosis of NGOC, although occurrence of first intercourse should not necessarily exclude NGOC. Once there is suspicion for a NGOC, a CT scan should be obtained looking for metastatic disease. Metastatic disease may be seen in the lungs, pelvis, vagina, and the liver. Metastatic disease often results in hemorrhagic lesions which can rapidly become life threatening. The CT scan should include evaluation of the head, chest, abdomen, and pelvis.

Surgical resection is required for improvement in both overall and progression free survival. However, full surgical staging may be more morbid than is required to effectively treat patients with NGOC and may compromise fertility in younger patients. Combined with the concern that many of these patients are pre-pubertal or of reproductive age, fertility-sparing options may be reasonable after shared decision making with the patient. Patient should be typed and crossed when possible in order to provide crossmatched blood. Many case series describe patients with life-threatening low hemoglobin levels due to the exquisitely hemorrhagic nature of NGOC. Extra-abdominal sites of hemorrhage should also be considered in profoundly anemia individuals when there is absence of free fluid in the abdomen or pelvis or lack of vaginal bleeding. In the rare circumstance that a patient will not accept blood, alternative products, cell saver, or other blood recycling devices should be on hand and ready to be used if amenable to the patient. If not, then it should be clearly documented that the patient understands risks related to this refusal. Minimally invasive surgery can also be considered given its association with shorter hospital stay and lower blood loss, but the risks of port-site recurrence or inadequate staging and/or cytoreduction should be factored into the decision to proceed laparoscopically. 

NGOC is more closely related to germ cell tumors and as such should be treated with platinum-based chemotherapy. Single agent choriocarcinoma treatments are ineffective in these cases. The treatment most used in case reports was BEP, although remission has also successfully been achieved with EMA/EMA-CO. Three cycles of BEP appear to be adequate for patients that had localized disease; however, with more advanced or bulky disease, four cycles may be used. When fertility-sparing surgery has been used, the gonadotoxic effects of chemotherapy should also be discussed with the patient. 

Close monitoring should be in place to allow for early recognition and urgent treatment of rapidly fatal disease complications such as a ovarian hyperstimulation syndrome and choriocarcinoma syndrome is imperative. Multimodal supportive therapy should be initiated promptly upon recognition. When appropriate, lower-dose chemotherapeutic regimens may be considered if a patient is thought to be at risk for developing choriocarcinoma syndrome. 

Long-term surveillance is necessary for these patients to detect disease recurrence. Liu et al. provided a schema for surveillance as described above in Section 6. These recommendations are provided as a guide based on the available data regarding NGOC and may be considered when treating a patient with this extremely rare disease.

## 7. Conclusions

NGOC is a distinct and rare disease from the more common GOC and poses diagnostic challenges as its presentation can mimic gestational trophoblastic disease or more common conditions in reproductive-aged women (i.e., ectopic pregnancy) and cannot be differentiated from GOC on histopathology. Therefore, special attention needs to be paid to ensuring the prompt and proper diagnosis by acknowledging a patient’s history that excludes or minimizes the possibility of pregnancy and utilizing tissue genotyping (when appropriate). Patients presenting with a large, unilateral solid tumor should raise the suspicion of a germ cell tumor and negative point-of-care pregnancy tests should be confirmed with a serum beta HCG level. With proper diagnosis, chemotherapy, and surgical resection, patients can experience positive outcomes both from a survival benefit as well as fertility preservation.

## Figures and Tables

**Table 1 diagnostics-12-00560-t001:** Features of NGOC, GOC, and ectopic pregnancy.

	NGOC	GOC	Ectopic Pregnancy
Chorionic villi	No	No	Yes
Abnormal trophoblastic tissue	Yes	Yes	No
Abdominal pain and vaginal bleeding	Yes	Yes	Yes
Positive pregnancy test	Yes	Yes	Yes
Paternal genetic material	No	Yes	Yes
Treatment	Surgery/chemotherapy	Chemotherapy	Surgery and/or methotrexate

**Table 2 diagnostics-12-00560-t002:** List of cases of Non-Gestational Ovarian Choriocarcinoma.

Author	Age (Years)	Beta-HCG at Diagnosis (mIU/mL)	Surgical Treatment	Chemotherapy Treatment	Outcome/Follow-Up (Months) *
Peng [47]	16	120,420	USO	Actinomycin & Etoposide --> EMA-CO	Survived/3 DF
Adow [14]	25	1,000,000	HYST w/ BSO	BEP	Survived/12 DF
Heo [4]	12	20,257	Left USO	BEP	Survival/14 DF
Yee [26]	16	624,177	Left ovarian cystectomy w/partial oophorectomy	BEP	Died
Kumar [22]	34	877,414	No surgical resection	Radiation --> BEP--> Vinblastine, Ifosfamide, Cisplatin	Survived/6 DF
Goyal [48]	18	3751	Right USO	BEP	Survived/6 DF
Syed [24]	38	300,000	Left USO w/omentectomy	BEP	Survived/UNK
Rao [1]	26	8160	Right USO/partial omentectomy/partial splenectomy/right adrenalectomy	BEP	Survived/UNK
Yamamoto [1,49]	19	206,949	Left oophorectomy	EMA	Survived/12
Balat [1,25]	24	8968	HYST w/BSO, partial omentectomy and sternum mass excision	BEP	Died
Byeun [1,50]	28	13,378	Right USO	EMA	Survived/UNK
Corakci [1,51]	22	15,050	HYST w/BSO and partial LND	BEP	Survived/12 DF
Lyn [1,52]	48	7663	HYST w/BSO, partial LND, omentectomy, appendectomy, and peritoneal biopsy	BEP	Survived/12 DF
Park [53]	55	64,838	HYST w/BSO	BEP	Survived/20 DF
Nishino [11,38]	38	5030	HYST w/BSO, left lung segmentectomy	EMA, paclitaxel and cisplatin, fluorouracil and actinomycin-D, EMA-CO	Died
Hayashi [54]	10	6600	R USO	BEP	Survived/62 DF
Yang [11]	14	764,826	R USO --> HYST w/omentectomy	EMA-CO --> vincristine, actinomycin-D, etoposide, fluorouracil	Survived/12 DF
Xin [55]	23	18,000	L cystectomy --> L USO, omentectomy, peritoneal biopsy, retroperitoneal LND	BEP	Survived/9 DF
Choi [56]	33	74,612	L USO, peritoneal biopsies, R cystectomy; endometrial biopsy	EMA	Survived/60 DF
Gerson [27]	33	564,000	R USO --> HYST, L USO --> splenectomy	EMA-CO	Survived/12 DF
Roghaei [57]	47	970	HYST, BSO, pelvic LND, partial omentectomy	EMA-CO, vincristine	Survived/UNK
Irene [58]	9	444,900	HYST, BSO, partial omentectomy, appendectomy	EMA, cyclophosphamide, vincristine	Survived/UNK

* At time of article publication; USO = unilateral salpingo-oophorectomy; BSO = bilateral salpingo-oophorectomy; LND = lymph node dissection; HYST = hysterectomy; BEP = bleomycin, etoposide, and cisplatin; EMA = etoposide, methotrexate, actinomycin D; EMA-CO = etoposide, methotrexate, actinomycin D, cyclophosphamide, and vincristine; DF = disease free, UNK = unknown.

**Table 3 diagnostics-12-00560-t003:** Case Series Treatment Protocols.

**Liu et al. [23] Treatment Protocols**	
**Patients**	**Chemotherapy**
14	Bleomycin, Etoposide, and Platinum
3	Etoposide, Methotrexate, Actinomycin D, Cyclophosphamide, and Vincristine
2	Methotrexate
1	Methotrexate + Cyclophosphamide
2	Vincristine + Cisplatin and Paclitaxel + Cisplatin
1	Cisplatin + Bleomycin + Cyclophosphamide
**Shao et al. [37] Treatment Protocols**	
20	Etoposide, Methotrexate, Actinomycin D, Cyclophosphamide, and Vincristine
17	Floxuridine, Actinomycin-D, Etoposide, and Vincristine
7	Bleomycin, Etoposide, and Platinum
4	Bleomycin, Vincristine, and Cisplatin
2	Ifosfamide, Carboplatin, and Etoposide

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
