# Peer review of "Non-Gestational Ovarian Choriocarcinoma: A Rare Ovarian Cancer Subtype"

_diagnostics, 2022, doi:10.3390/diagnostics12030560_

Round 1

Reviewer 1 Report

  1. This is an excellent review of a rare but interesting clinical problem.
  2. Line 17: suggest adding paternal "genetic" components.
  3. Line 33: suggest using similar metrics for GOC (1:369,000,000) and NGOC (0.6%) for easier comparison, if possible.
  4. Multiple references to stage of presentation and survival rates may be confusing to the reader. line 44: relatively poor prognosis; line 46: 100% 3-year survival for stage 1-III; line 126: >80% had metastatic disease ie stage 4; line 201: 3-year survival 92% with chemotherapy followed by complete cytoreduction etc
  5. lines 29-49: I feel the first paragraph of the introduction should be divided into 3 paragraphs. Para 1 on GOC: lines 29-37. Para 2 on NGOC: lines 37-43. Para 3: brief summary of % patients presenting at different stages and range of survival. This is then further explained in the text.
  6. Line 300: may be worth stating that serum HCG is rarely greater than 100,000 in ectopic pregnancy and should raise concern about an alternative diagnosis.  
  7. The recommendations are in line with the literature review. The majority of patients can be treated with partial surgical staging combined with fertility-preserving surgery and chemotherapy, in the context of shared decision making with the patient. 

Author Response

2/5/2022
Dear Editor

Thank you for taking the time to review our article and provide feedback.  I have included a detailed description of our responses.
1. I agree, adding genetic will improve clarity and we have updated the manuscript.
2. The metrics for GOC and NGOC are difficult to calculate.  The reason that we used this terminology is that is how it has been reported in other case reports.  I did update the verbiage to make it more clear.  The reported percentage of NGOC is less than 0.6% of malignant germ cell tumors.  I was unable to find an incidence for NGOC.  However, the evidence is that it is less common than GOC.  However, if I were to calculate the incidence using available data for germ cell tumors the incidence would be around 1:15,000,000 which is higher than GOC, which I think would add to confusion.  I feel that it is best to leave it the way it currently is.  We could alternately just leave the 0.6% out and say less likely.  Please let me know if you feel that this would be a better option?
3.  The differences are related to different variables.  In the first we are talking about survival at different stages FIGO for Choriocarcinoma and Ovarian Cancer.  We have changed the wording to make it more clear for the readers.
4. Thank you for this input, we changed the opening paragraph as you have suggested
5. We have added this to clarify the fact that severely elevated beta-HCG is usually not seen in ectopic pregnancy.
Again, thank you for your time and input.

Respectfully 
Sean Cronin MD

Reviewer 2 Report

This study is comprehensive and well set. The study data are of interest and valuable. Overall, the manuscript sections are questionable, and some sections are in excess; thus, the whole manuscript should be rearranged upon the right sections.

Background:

Table 1. is probably misnamed.

Presentation:

Page 45, line 146: the sentence “In a review article looking at 37 patients with NGOC, stage was calculated using both the 2013 FIGO staging for ovarian cancer and the 2000 FIGO staging for choriocarcinoma” is missing author name … In a review article of XY author… The whole paragraph isn’t clear enough regard the literature data.

Diagnosis:

The whole chapter should be re-written with the better linking of sentences/findings and citing cause-and-effect relationships.

Treatment:

Table 2. contains very important and valuable data. Well done! Table 3. is missing results of treatments for Shao protocols. If they are not available, it should be considered to move this part of table and possibly show the data in the form of text.

Author Response

2/5/2022

Dear Editor

Thank you for taking the time to review our article and provide feedback.  I have included a detailed description of our response
1. Table 1 was renamed to better reflect its information as a comparison of different conditions.
2. We have added the authors names to better reflect their input.
3. Thank you for the input regarding the diagnosis and presentation section.  We have combined these two sections into one section, removed much of the redundant data we believe that it reads much better in its current form.  Please let us know if you think of the new organization.
4. Thank you for the compliment on table 2, Dr. Ahmed and I spent a lot of time finding the available data.  With respect to table 3, the goal was to show the treatment modalities.  The Liu article did report on the outcomes for each treatment group, but the Shao article did not.  Since we agree that including it for one and not the other looked odd. We removed the Liu data as it was not the intent of this table.  Let us know if you agree with this change.
Again, thank you for your time and input.

Respectfully 
Sean Cronin MD